# Evolution of flood generating processes under climate change in France

Yves Tramblay [1], Guillaume Thirel [2,3], Laurent Strohmenger [2,4], Guillaume Evin [5], Lola Corre [6], Louis Heraut [7], Eric Sauquet [7]

[1] Espace-Dev, Univ. Montpellier, IRD, Montpellier, France

[2] University of Paris-Saclay, INRAE, HYCAR, Antony, France

[3] Univ Toulouse, CNES/IRD/CNRS/INRAE, CESBIO, Toulouse, France

[4] Department of Ecohydrology and Biogeochemistry, Leibniz Institute of Freshwater Ecology and Inland Fisheries (IGB), Berlin, Germany

[5] Univ. Grenoble Alpes, INRAE, CNRS, IRD, Grenoble INP, IGE, Grenoble, France

[6] CNRM, Université de Toulouse, Météo-France CNRS, Toulouse, France

[7] INRAE, UR RiverLy, Villeurbanne, France

*Correspondance to: Yves Tramblay (yves.tramblay@ird.fr)*

Revised manuscript, July 2025

**Abstract**

The impact of climate change on floods varies spatially, and often the observed trends in flood characteristics can be explained by differentiated changes in flood-generating processes. This study explores changes in flood magnitude and flood-generating processes under different climate change scenarios for a large number of basins in France. It is based on an unprecedented exercise to model the impacts of climate change on hydrology, using a semi-distributed model (GRSD) applied to 3727 basins with 22 Euro-CORDEX bias-corrected climate projections using two greenhouse gas emission scenarios (RCP4.5 and RCP8.5). Annual maxima of daily simulated streamflow were extracted for the period 1975-2100, resulting in a set of over 10 million flood events, and a trend analysis was carried out on both flood magnitudes and flood generating processes. Increasing trends in flood magnitudes are only found in the northern regions of France, although multi-model convergence rarely exceeds 60 %. The highest increases are observed for the 20-year floods and under the RCP8.5 scenario. A classification of floods according to their generating process revealed that floods linked to soil saturation represent more than half of all floods in France. The relative change in the importance of the different flood-generating processes is not spatially homogeneous and varies by region. The proportion of floods linked to soil saturation excess is increasing in the Northeast, while decreasing in the southern Mediterranean regions. In these Mediterranean regions, the proportion of floods linked to infiltration excess related to extreme rainfall is increasing. Both the frequency and magnitude of floods linked to snowmelt processes are decreasing in mountainous areas. On the contrary, the most extreme floods associated with rainfall on dry soils tend to increase, in line with the increase of rainfall intensity. Overall, trends in antecedent soil moisture conditions are as important as trends in intense rainfall to explain flood hazard trends in the different climate projections. This study shows how important it is to decipher the changes in the different flood generating processes in order to better understand their evolution in different hydroclimatic regions.

**1- Introduction**

The impact of climate change on floods is uncertain, or unknown, in many regions of the world (Intergovernmental Panel On Climate Change (IPCC), 2023). In particular, it has been shown that competing changes in flood-generating processes can modulate, or even offset, changes in flood hazards (Ivancic and Shaw, 2015; Sharma et al., 2018; Tramblay et al., 2019; Brunner et al., 2021; Zhang et al., 2022). For example, in the context of an increase in intense rainfall events in some temperate and arid regions, the concomitant drop in soil moisture, resulting in lower runoff coefficients, could result in a lack of trend in flood magnitude (Wasko and Nathan, 2019; Ho et al., 2022; Tramblay et al., 2023; Scussolini et al., 2023). Similarly, the large decline in the frequency of snowmelt-induced floods in many regions may compensate for the increasing proportion of floods caused by rainstorms, resulting in the absence of trends in overall flood hazards in numerous regions worldwide where snowmelt is prevalent (Zhang et al., 2022). To comprehend the evolution of floods, it is essential to analyze not only their severity but also to consider in detail whether the processes that underpin them are also changing (Tarasova et al., 2019; Kemter et al., 2020; Blöschl, 2022a, b; Jiang et al., 2022; Tarasova et al., 2023; Kemter et al., 2023; McMillan et al., 2025). This is particularly salient given that trends in extreme rainfall do not generally translate into the same trends in floods (Ivancic and Shaw, 2015; Sharma et al., 2018; Wasko et al., 2023). Numerous studies have focused on understanding and categorizing floods according to the hydroclimatic mechanisms

behind their generation (Tarasova et al., 2019; Stein et al., 2020; Kemter et al., 2023). We can distinguish studies that use seasonality to categorize the various causal mechanisms based on their temporal proximity (Berghuijs et al., 2016, 2019; Tramblay et al., 2021) from other studies that use different hydrological criteria, predominantly thresholds on precipitation intensity, snowmelt, or soil moisture levels, to differentiate between different types of floods (Tarasova et al., 2020; Stein et al., 2020, 2021; Tramblay et al., 2022; Tarasova et al., 2023; Tramblay et al., 2023).

To provide future flood projections, it is necessary to use hydrological models to translate climate projections into hydrological projections, i.e. providing simulated streamflow values for the future. Such a modeling chain, which is generally constituted by greenhouse gas concentration scenarios, global (GCM) and regional (RCM) climate models, bias-correction methods, and hydrological models, necessarily suffers from uncertainties (Clark et al., 2016). The evolution of our society is inherently unpredictable, and thus, climate scenarios describe its evolution through greenhouse gas concentrations (Meinshausen et al., 2020). These concentrations, in turn, influence climate model outputs, which are subject to uncertainties related to process representation and climate variability (Knutti and Sedláček, 2013). Additionally, bias-correction methods and hydrological models are affected by process representation uncertainties (Teutschbein and Seibert, 2012; Maraun et al., 2017). Furthermore, their application in an extrapolation mode may compromise their temporal transferability. Indeed, these models and methods are generally calibrated, optimized, and evaluated over well-known past periods. With climate projections, they are used under potentially very different conditions, with unprecedented air temperatures, and possibly very different precipitation regimes, which may alter their transferability. This issue has been extensively documented in the context of hydrological modeling (Brigode et al., 2013; Thirel et al., 2015b; Dakhlaoui et al., 2017), and various protocols have been proposed to assess the robustness of these models (Klemeš, 1986; Thirel et al., 2015a). It is important to note that models of varying complexity exist, depending on their spatial discretization and the processes included in the equations, as well as the use of automatic parameter optimization (Hrachowitz and Clark, 2017). For large-scale studies, at the regional or even the global scale, most often the hydrological models are not calibrated for all stations but sometimes only in a subset of basins (Alfieri et al., 2015; Roudier et al., 2016; Do et al., 2020; Di Sante et al., 2021; Zhang et al., 2022). Nevertheless, all types of models can suffer from robustness issues (Santos et al., 2025). However, it has been shown that the relative uncertainty associated with hydrological modeling is typically lower for high flows, which are the focus of this study, in comparison to low flows (Vidal et al., 2016; Lemaitre-Basset et al., 2021).

This study analyzes a large ensemble of hydrological projections based on climate projections for a diverse set of basins in France, which represents a range of varied physiographic properties. The aim is to investigate the future evolution of flood events, both in terms of magnitude and flood-generating processes. In a multi-model context, it is important to analyze both the convergence of the scenarios for the different models and the signal of change given by these different models. The methodology enables a multi-factorial analysis of floods and the processes underlying their generation, leading to a better understanding of which factors most influence flood trends. By distinguishing the hydro-climatic drivers regionally, we identify those that may either increase or decrease future flood risks.

**2- Data and Methods**
**2.1 Hydrological model**
The GRSD model, a semi-distributed rainfall-runoff model based on the lumped GR4J model
structure (Perrin et al., 2003) and the CemaNeige snow model (Valéry et al., 2014), has been
chosen for this study. The rationale behind this choice is its capacity to be applied to a very
large range of hydro-meteorological conditions and numerous basins across the world, as
shown by the GR4J or GRSD applications in Brazil (Kuana et al., 2024), Nepal (Nepal et al.,
2017), Iran (Jahanshahi et al., 2025), Haiti (Bathelemy et al., 2024), Chile (Abbenante et al.,
2024), Australia (Stephens et al., 2019), and France (De Lavenne et al., 2019; Lemaitre-
Basset et al., 2024), as well as its large number of applications in the context of climate change
(Chauveau et al., 2013; Cornelissen et al., 2013; Tian et al., 2013; Fabre et al., 2016; Stephens
et al., 2018; Givati et al., 2019; Thirel et al., 2019; Tarek et al., 2021; Wasko et al., 2023;
Poncet et al., 2024). The GRSD model calibration and simulation were applied within the
airGR and airGRiwrm open-source R packages (Coron et al., 2017; Dorchies, 2022).
GR4J is a four-parameter rainfall-runoff model consisting of two conceptual stores: a
production store and a routing store, whose flows are routed to the river after transformation
by a unit hydrograph. While the GR4J model is applied on a sub-catchment spatial scale, and
the simulated streamflow is routed from upstream to downstream with a lag function (de
Lavenne et al., 2019), the CemaNeige snow model uses an additional sub-division of each
sub-catchment into five zones of equal area to better describe the process heterogeneities
linked to the topography (Valéry et al., 2014). GRSD provides daily streamflow simulation
covering 3727 simulation points in France, and different variables have been extracted: the
total, solid, and liquid precipitation, the soil water index (SWI, i.e. the ratio between water
content and the storage capacity of the production store), and the snowmelt discharge (SMD,
i.e the water flow resulting from snowpack melting). The GRSD simulations used in the present
study are described in Sauquet et al., (2024, 2025). The GRSD model has been calibrated
over the 1976-2019 period, using the Kling-Gupta Efficiency criterion (Gupta et al., 2009)
applied on squared-root transformed streamflow, against the observed streamflow of 611
gauging stations in France considered as nearly natural (Strohmenger et al., 2023), using the
SAFRAN (Vidal et al., 2010) reanalysis as meteorological input. To simulate streamflow over
the remaining - ungauged - stations, parameters were transferred from neighbor gauged
catchments to the target ungauged catchments following Oudin et al. (2008), and these
parameter sets were used to produce pseudo-observed streamflow on the ungauged stations,
on which the GRSD model was subsequently calibrated (Sauquet et al., 2024, 2025). The
validation of the GRSD model is presented in Sauquet et al., 2024, 2025 and Héraut et al.,
2024. In addition, the comparison of the observed and simulated 2-year and 20-year floods is
presented in the supplementary materials, figure S7, showing a relative bias of -3% and mean
absolute relative error of 16% for 2-year floods, and 20% for 20-year floods.
**2.2 Climate simulations**
The GRSD model was thereafter applied using as inputs climate projections from the
Explore2-Climat-2022-ADAMONT dataset (Corre et al., 2025), available on the French
national climate data portal (*DRIAS - les futurs du climat* - www.drias-climat.fr), and described
in Sauquet et al. (2025). This ensemble is derived from the 12-km resolution EURO-CORDEX
ensemble (Jacob et al., 2018; Vautard et al., 2021), which consists of regional climate model
simulations that downscale global climate model simulations over Europe from the CMIP5 (5th
phase of the Coupled Model Intercomparison Project, (Taylor et al., 2012)). For this study, we
use the 11 GCM/RCM pairs for which both emission scenarios (RCP4.5 and RCP8.5) are
available (Table T1 in supplementary materials). The outputs of the climate models were
statistically adjusted over mainland France, against the 8- km resolution SAFRAN reanalysis
(Vidal et al. 2010). The bias correction method consists of a quantile mapping relying on
seasonal weather regimes (ADAMONT method, Verfaillie et al., 2017). In the following, we
use the daily rainfall data and the variables used to compute evapotranspiration (temperature,
relative humidity, wind, radiation) from these bias adjusted simulations. These data are
available from 1975-2100 at the daily time scale with an 8-km spatial resolution and have been
aggregated at the catchment scale.
Several studies describe the future climate changes in France based on these projections
(Marson et al., 2024). Overall, they show a shift towards hotter and drier summers, as well as
an expansion of temperate climate in mountainous regions (Strohmenger et al., 2024). Despite
variations in the magnitude of warming depending on the scenario, time horizon, or level of
warming considered, all studies identify common features. These include spatial contrasts,
with stronger warming projected in the south, east, and mountainous regions compared to the
Atlantic coast, showing a gradient of about 1°C. Seasonal contrasts are also consistent, with
warming more pronounced in summer than in winter (with also a difference of 1°C). Regarding
precipitation, all studies highlight the limited agreement among models on the annual average,
which is linked to the influence of internal variability and to France's geographical position,
which places it in a transition zone between Northern Europe, where precipitation is projected
to increase, and the Mediterranean region, where it is projected to decrease (Terray and Boé,
2013; Coppola et al., 2021). On a seasonal scale, however, projections are more consistent.
In winter, median changes show an increase in rainfall (particularly in the northern regions)
alongside a notable decrease in snowfall, especially in the low- and mid-mountain ranges. In
summer, projections indicate a decrease in rainfall (particularly in the southern half). These
changes in precipitation combined with increased evaporation due to warming, lead to a
decrease in water resources availability and, consequently, a marked drying of the soil. In
terms of intense rainfall (characterized by the annual maximum daily rainfall), the climate
models show large dispersion regarding the direction of future changes (Tramblay et al.,
2024), except for the long-term horizon under the RCP8.5 scenario or for high levels of
warming (e.g., +4°C on average across France), where they tend to agree on an increase,
particularly in the northern part of France
**2.3 Extraction of flood events**
From this dataset, we focused on the annual maximum of daily streamflow that has been
extracted for each simulation point. For each station, a time series of 124 values, between
1976 and 2099 considering hydrological years starting in September, is available and
described thereafter as the Annual Maximum Flood (AMF) series, referred to as 'floods' to
improve readability. This sampling results in a total of 10,167,256 flood events. To estimate
the base flow contribution, that is the proportion of direct runoff from rainfall and/or snowmelt,
we applied a base flow filter (Lyne and Hollick, 1979) to compute the direct streamflow fraction
from the full daily times series, to be able to compute the base flow contribution for each flood
event. To extract the characteristics of the events in terms of rainfall and snowmelt, we
computed the antecedent cumulative rainfall (in mm) and snowmelt (in mm) for each AMF
value, and this aggregation stops if a day has rainfall below 1 mm or if a maximum of seven
days is reached. Thus, this duration estimated for each event is extracted and hereafter
referred to as the time of concentration, which may differ for each event. This time window of
seven days is chosen given the size of the basins considered in the present study (from 64 to
111 570 km²) and also to be consistent with previous studies using similar approaches (Ivancic
and Shaw, 2015; Stein et al., 2020). For each simulated flood event, we extracted: the
concentration time (in days), the antecedent soil moisture (i.e. SWI, between zero and one)
one day before the concentration time, the total and maximum daily rainfall (mm) during the
event, the fraction of the total flood streamflow being direct streamflow (%), and the total
snowmelt discharge (mm) during the event.
**2.4 Classification of flood events**
The flood event classification was determined using a decision tree that had previously been
applied in Europe and the USA (Kemter et al., 2020, 2023) to estimate the importance of five
different flood-generating processes in each catchment, namely long rain, short rain,
snowmelt, rain and snowmelt, and soil moisture excess. This classification is based on the
catchment and climatic conditions occurring during the period defined by the time of
concentration before the day of the flood peak. The rationale behind the choice of this
classification over others (e.g. Tarasova et al., 2019) is that it does not rely on fixed quantities
for the different variables, and notably precipitation extremes, compared to other classification
schemes (Stein et al., 2020; Tramblay et al., 2022, 2023). Indeed, in a climate change context,
these quantities could vary over time, notably those related to extreme rainfall that are
inherently difficult to estimate precisely using climate simulations at these spatial and temporal
resolutions (8 km, daily).
The flood event classification from Kemter et al. (2023) is adapted using the following
sequential rules:
1. If the snowmelt is larger than the rainfall, the event is classified as 'snowmelt'.
2. If the snowmelt is larger than 25 % of the rainfall, the event is classified as 'rain and
272        snowmelt'.
3. If antecedent soil moisture before the event is larger than the soil moisture threshold,
274        the event is classified as 'soil water excess'. These events correspond to soil saturation
275        excess.
4. If during the event, the daily maximum rainfall exceeds 75 % of the total rainfall, the
277        event is classified as 'short rain'. These events are representing soil infiltration excess
278        caused by a short and intense rainfall event.
5. Else, the event is classified as 'long rain'.
In the present study, we estimated the soil moisture threshold for each basin to identify flood
events related to soil moisture excess. Considering specific soil moisture thresholds for each
basin is undoubtedly more realistic than using a single soil saturation threshold for all basins,
as done in other studies. The approach used herein has been applied previously (Tarasova et
al., 2018; Tramblay et al., 2022) and is data-based, since there is no method for estimating
these thresholds a priori from catchment characteristics. It relies on the extraction of all
streamflow events above the 10th percentile of the whole streamflow time series between
1975 and 2005, fitting an exponential function to the soil moisture/runoff relationship, and then
to identify the inflection point in the soil moisture/runoff relationship (i.e., the max slope of the
curve). Here, we apply the Pruned Exact Linear Time method (Killick et al., 2012) to detect a
change point (i.e., the soil moisture threshold, figure S1) in the exponential fit.

**2.5 Trend detection**

To check for trends, the entire annual time series between 1976 and 2099 was considered.
The rationale for using trend detection over the whole record is that floods in many regions
are strongly impacted by multidecadal variability, notably in Europe (Hodgkins et al., 2017;
Lun et al., 2020; Blöschl et al., 2020), making it difficult to detect climate change response
using the standard approach of comparing historical and future time windows of about 30
years. Two approaches are applied to detect these trends. First, a quantile regression
(Koenker and Bassett, 1978), is applied to check the presence of trends in the 2-year flood
and in the 20-year flood. The significance of these trends at the 5 % level is then assessed
using a bootstrap method (Efron, 1979). In this case, the magnitude of the trend is given by
the slope of the regression. We also applied a variant of the Mann-Kendall test adjusted for
serial correlation in the data (Hamed and Ramachandra Rao, 1998), and the Sen slope (Sen,
1968) method to estimate the magnitude of the trends. The Mann-Kendall test was applied to
the different flood event characteristics (event rainfall, antecedent soil moisture, direct runoff
fraction, snowmelt contribution), as well as the relative frequencies of the different flood event
types. In that latter case, 20-year sliding windows were used to compute these frequencies,
using a similar approach as in Tarasova et al. (2023). Given that some processes are not
relevant to all basins (e.g. snowmelt), we applied the trend detection test only if a minimum of
20 occurrences for the given flood process is present in the data sample over the 1975-2100
period. To present the trend detection results, we considered the multi-model index of
agreement (MIA) (Tramblay and Somot, 2018) that describes the model convergence towards
an increase (1) or a decrease (-1) for a given indicator. The objective is to identify robust
changes, where all model projections converge towards the same result. Finally, the trends in
the flood quantiles (Q50 and Q95) and the trends in the five flood-generating processes have
been clustered for the two RCP separately, using the Ward linkage method, commonly used
in hierarchical clustering, together with silhouette plots to identify the optimal number of
clusters (Kaufman and Rousseeuw, 1990). This, in order to define regions with similar flood
changes.

**3- Results and discussion**


**3.1 Changes in flood characteristics**


Flood trends are analyzed here using the quantile regression technique, which distinguishes
the change signal for AMF corresponding to 2-year (Q50) and 20-year (Q95) return periods.
The results presented in Figure 1 show the convergence of the different simulations under the
RCP4.5 and RCP8.5 scenarios with the MIA index. The figure shows a contrasting signal, with
a relatively good convergence of the models towards an increase in Q50 and Q95 in the
northern half of France and an absence of consensus for trends or even a decrease in some
areas for the southern regions, that is more marked under RCP8.5. Overall, convergence at
the country level is rather weak, with the median MIA value around 0.5 (Figure S2).
Convergence between models is slightly greater for Q50 rather than Q95, expressing a larger
uncertainty for rarest floods. In terms of magnitude (Figure 2), trends are more marked for the
rarest floods (median change +28 % with RCP4.5, +34 % with RCP8.5), than for the most
frequent floods (median change +20 % with RCP4.5, +27 % with RCP8.5). Overall, the
projected changes between Q50 and Q95 are more correlated under the RCP8.5 than under
the RCP4.5. This north/south contrast in the projections is consistent with previous findings
using observational data (Blöschl et al., 2019; Tramblay et al., 2019).

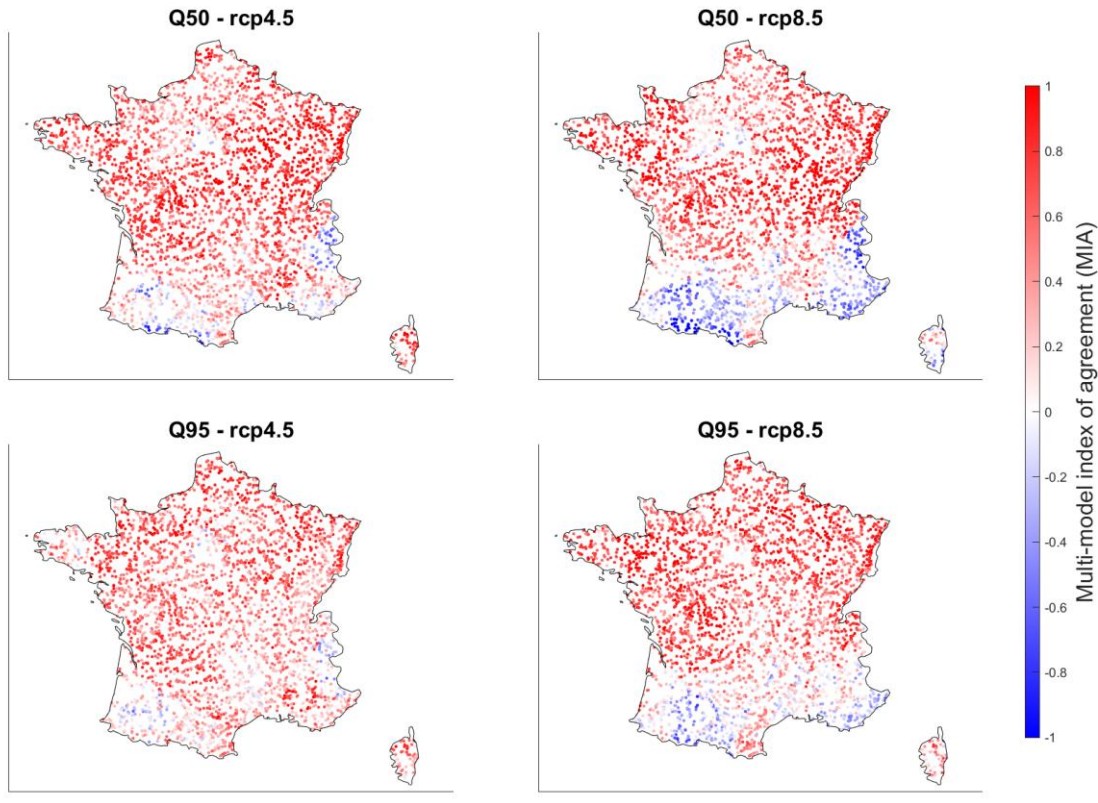

Figure 1: Maps of the multi-model index of agreement for the Q50 (2-year flood) and Q95
(20-year flood) for the 3727 simulation points, for the RCP4.5 and RCP8.5. The multi-model
index of agreement (MIA) is equal to 1 if all models project a significant increase, and equal
to -1 if all models project a significant decrease.

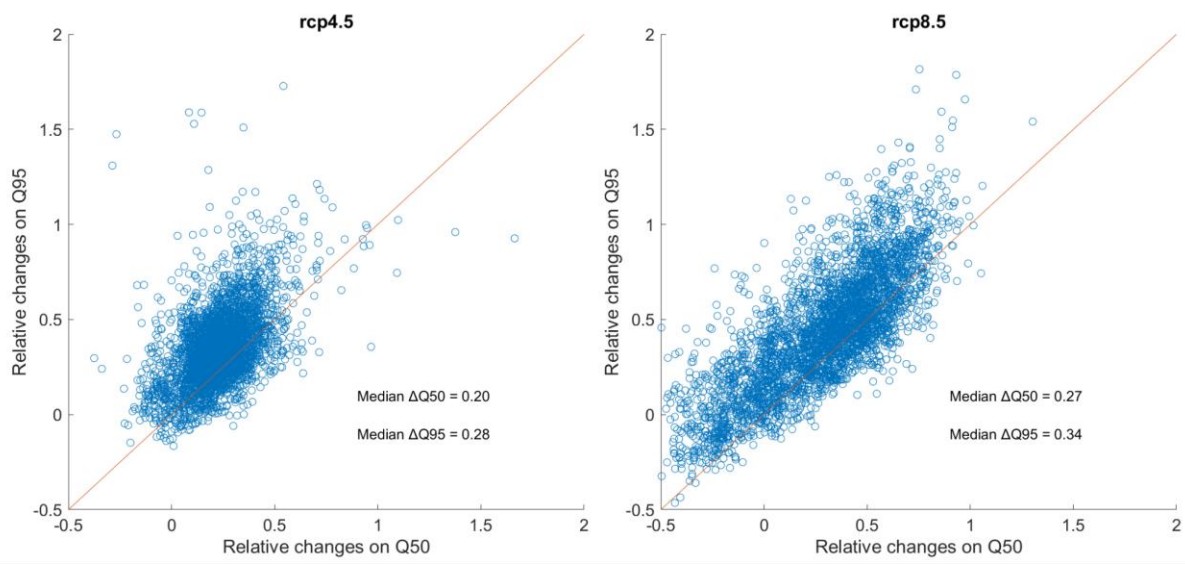

Figure 2: Comparison of the mean multi-model magnitude of significant trends on 2-year (Q50) and 20-year (Q95) floods under the RCP4.5 and RCP8.5.

This analysis of flood trends is complemented by trends in various indicators calculated during floods: maximum daily rainfall during the episode, initial soil moisture conditions, fraction of flood discharge from direct runoff, and contribution of snowmelt to flooding. The trends are shown for the RCP4.5 and RCP8.5 scenarios in Figure 3, with a general increase in maximum daily rainfall during floods, though less marked in southern France. The trends in antecedent soil moisture are more contrasted, with a good convergence of models towards an increase, more marked for RCP8.5, in the northern half of France and particularly in the eastern region. On the opposite, antecedent soil moisture decreases in the mountainous regions of the Alps and Pyrenees, and around the Mediterranean Sea. For the fraction of direct runoff (that is the total runoff minus the base flow for the day of the flood event), there is a slight trend towards an increase in RCP4.5, and a noticeable trend towards an increase in RCP8.5, in the western regions. Finally, the contribution of snowmelt to flood flows is decreasing wherever this component influences floods. Thus, it is interesting to note that in a context of increasing rainfall intensity, outside Mediterranean regions, we observe an increase in flood magnitude, that is associated either with an increase in initial soil saturation conditions, in eastern France, or with an increase in the fraction of direct runoff, in western France. It is also worth noticing that the spatial pattern of antecedent soil moisture trends seen in Figure 3 seems to mimic flood trends more closely than extreme rainfall trends, as observed in other regions of the world (Wasko and Nathan, 2019).

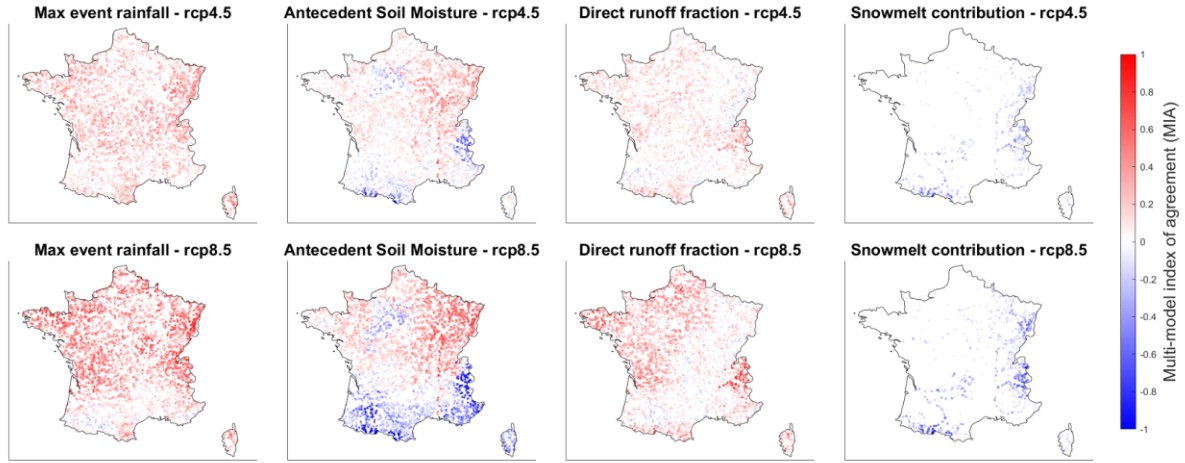

Figure 3: Multi-model index of agreement of the trends in maximum rainfall during floods, antecedent soil moisture conditions, direct streamflow fraction, and snowmelt fraction, under the RCP4.5 and RCP8.5 with ADAMONT

**3.2 Classification of flood-generating processes**

The floods extracted from the various simulations were classified according to the different categories; snowmelt, rain and snowmelt, soil water excess, short rain, and long rain. The result of this classification from the multimodel ensemble during the historical period 1975-2005 is presented in Figure 4, which shows the relative contribution of these different flood types for each station. Events linked to soil saturation are predominant, accounting for more than 50 % of floods in France. Long precipitation events are responsible for almost 25 % of floods and are mainly located in the north-west and south of France. It should be noted that most of these events also correspond to soil saturation processes, although in this case, the soil saturation occurs not before but during these events. Indeed, Figure S3 shows that during these long rain events, the largest floods are only associated with SWI values above 0.6, and that the modal value of SWI maxima is close to 0.75 and 0.8, corresponding to the value of the soil saturation threshold most frequently found in French basins (Figure S1). And, on average for all basins, the threshold that splits between long rain and soil water excess floods are exceeded 69% of the time during long rain floods. Snowmelt-related events, caused by snowmelt or a mixture of rain and snowmelt, account for around 11 % of the total number of floods, mainly for mountainous areas and central-eastern regions. Lastly, short rain events account for less than 10 % of the events, with a spatial distribution very similar to that of long rain. In terms of magnitude, when all the standardized floods for the different processes are grouped, short rain events show the highest magnitude, followed by soil water excess and long rain events, which have similar magnitudes (Figure 5a). This result is consistent with Tarasova et al. (2023) who found a more pronounced tail for the distribution of floods caused by intense rainfall on dry soils.

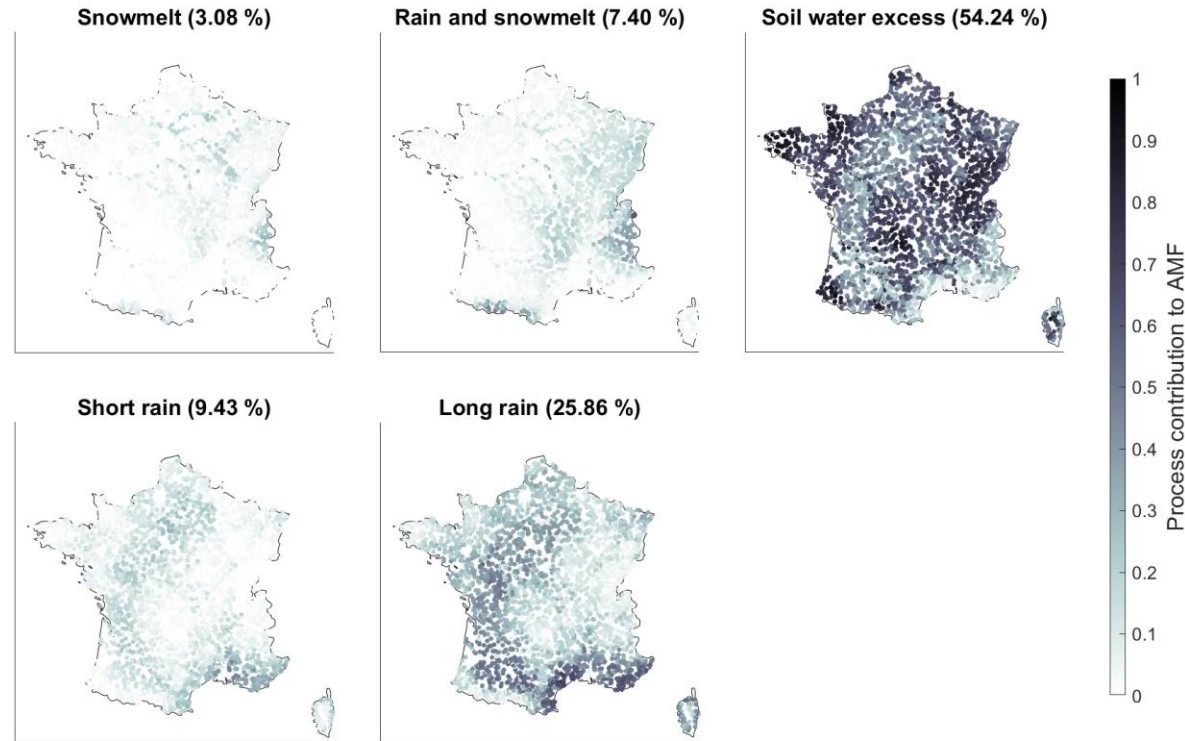

Figure 4: Relative proportion of the different flood-generating processes for each basin
during the historical period 1975-2005. The average contribution of each process to the total
411             number of floods is given in the titles of the sub-plots in percentages.
As these results are obtained with an ensemble of bias-corrected climate projections, the
results have also been extracted for each GCM/RCM couple independently to check the
consistency between results. Over the historical period, the classifications obtained with the
different climate models show a very high degree of consistency in the proportions of the
different classes between models (Figure S4). This result was quite expected given that the
RCM simulations are bias-corrected using the same reference. The results of this classification
are also very consistent with previous studies in Europe applying similar classification
schemes but with different datasets and a different methodology. For instance, Berghuijs et
al. (2019) have found that 49 % of floods are driven by soil water excess in Western Europe,
and about 22 % of floods are driven by maximum annual rainfall. Tarasova et al. (2023)
obtained similar results in their pan-European study using slightly different classes, with 11 %
of rain on dry soils, 67 % of rain on wet soils, and 21 % of rain-on-snow events in Atlantic
regions, and for the Mediterranean region, with 25 % of rain on dry soils, 59 % of rain on wet
soils, and 15 % of rain-on-snow events. In both regions, the proportion of snowmelt-only driven
floods was equivalent to 1 % only.

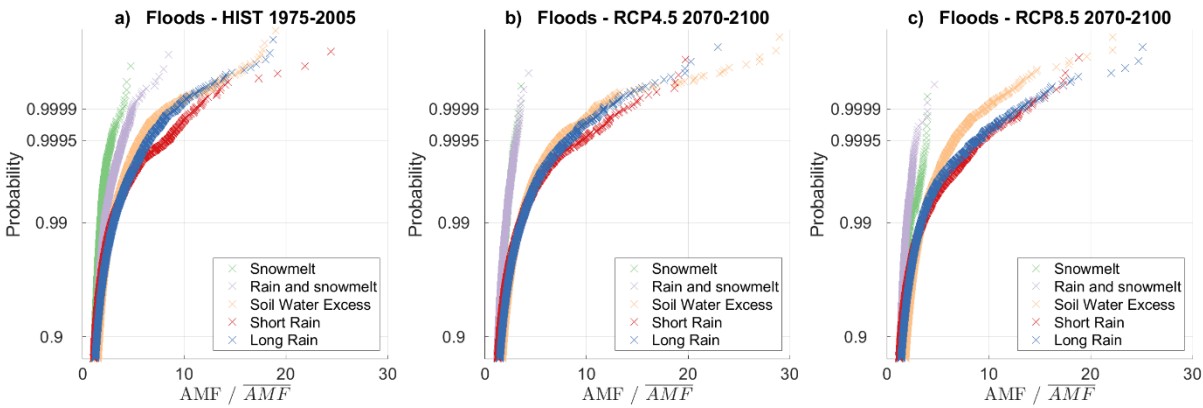

Figure 5: Distribution of floods all basins together, for the different types of floods during the historical period 1975-2005, and for 2070-2100 under RCP4.5 and RCP8.5. The annual flood (*AMF*) for each station have been standardized by the mean annual flood ($\overline{AMF}$)

## 3.3 Changes in flood-generating processes

The direction of changes in the contribution of the various flood-generating processes in RCP4.5 and RCP8.5 is shown in Figure 6. Given the low frequency of snowmelt-only events, it is difficult to draw robust conclusions about changes in these events. However, there is a general decrease in snowmelt or rain-and-snowmelt-induced floods in both scenarios. This applies both to the mountain ranges of the Alps and the Pyrenees and to regions located from the center of France to the northeast. There is also an increase in the contribution of events linked to soil saturation, especially in the northeast, that is more widespread spatially for RCP4.5 than for RCP8.5. Floods induced by short rains and long rains are increasing in areas where soil water excess events are decreasing, most importantly in the southern regions and the Alps. It should be noted that the change in rain and snowmelt events shows a very similar spatial distribution compared to the trends for events linked to soil water excess. Indeed, the spatial correlations between changes in soil water excess events on the one hand, and rain and snowmelt events on the other hand, are significant (-0.6 for RCP4.5 and -0.67 for RCP8.5). This means that where the proportion of events linked to the combination of rain and snowmelt is reducing, these events tend to be replaced by floods associated with saturated soils. Similarly, there are significant correlations between changes in the proportion of soil water excess events and the proportion of short rain or long rain events (correlations of -0.31 between changes in soil water excess and changes in short rain, -0.4 with changes in long rain under RCP4.5, and respectively -0.54 and -0.45 under RCP8.5 - yet the magnitude of these correlations remains small). We can therefore see that in regions where the proportion of floods related to saturated soils is decreasing, the proportion of floods associated with short rain or long rain is increasing, particularly in southern regions. This result shows that these shifts, previously observed in historical records (Jiang et al., 2022; Tarasova et al., 2023; Tramblay et al., 2023), are likely to amplify under the two emission scenarios considered herein.

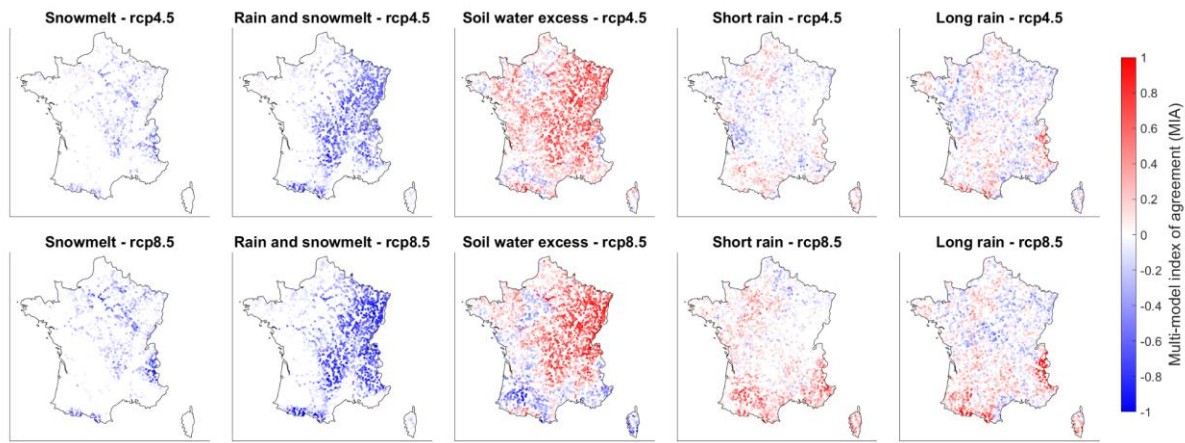

Figure 6: Multi-model index of agreement for the trends in the contributions of the different
flood-generating processes, under the RCP4.5 and RCP8.5
The classification of the trends in flood quantiles and the five flood generating process results
in four spatial clusters as shown in Figure 7. The first cluster (Mountain) includes mountainous
basins in the Alps and Pyrenees, where Q50 and Q95 show mostly declining trends, or an
absence of trends, linked to a sharp decrease in snowmelt-related events and an increase in
events caused by short rain and long rain. The second cluster (Mediterranean) includes basins
located in southern regions, where there is no trend or a decrease in floods, associated with
a drop in events linked to soil saturation and an increase in events caused by short rain and
long rain. The third and fourth clusters (Atlantic and Continental) gather the stations in the
northern half of France, where flooding trends are on the rise, but with a sharp decrease in
snowmelt-related flooding in the east (Continental cluster), while in the west (Atlantic cluster)
there is a slight increase of the frequency of intense rainfall (short rain) on dry soils, associated
with an increase in direct runoff, as already shown in the previous sections. The spatial
organization of the different clusters is very similar under the two scenarios RCP4.5 and
RCP8.5, with one notable difference concerning the Mediterranean cluster in the South of
France, which has a much more marked northward extension under the RCP8.5 scenario.
Overall, it should be noted that this spatial distribution is strongly reminiscent of the different
climatic zones observed in France (Strohmenger et al., 2024), with the western and coastal
regions under a temperate oceanic climate, the center and east of France with a more
continental climate, the southern part corresponding to areas with a Mediterranean climate,
and finally the mountain regions of the Alps and the Pyrenees.

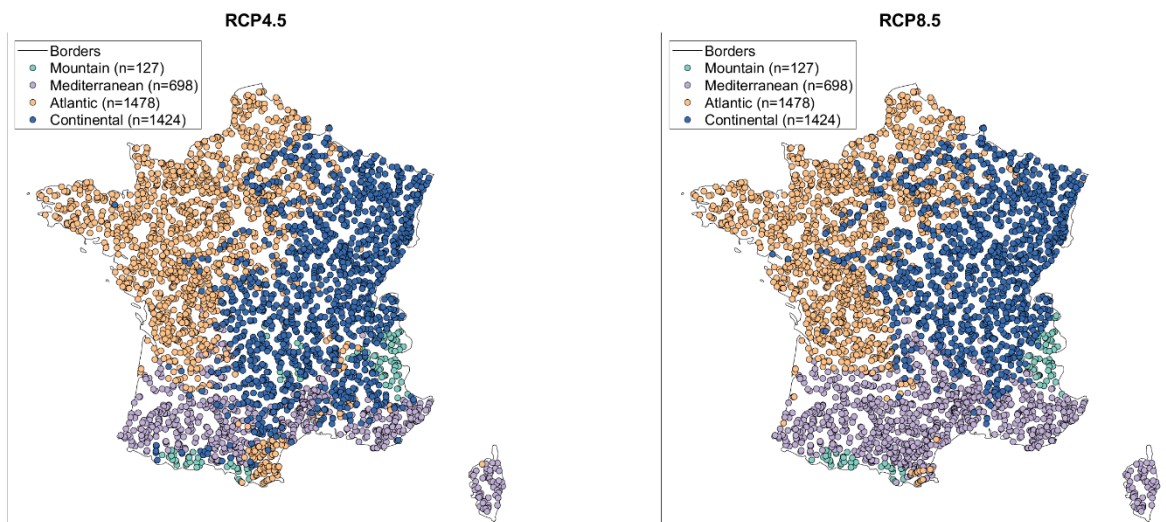


Figure 7: Regional clusters of the trends in flood magnitudes and flood-generating processes

**496 3.4 Regional drivers of change for flood hazard in future projections**


The magnitude of floods will change differently, depending on their generating processes.
When aggregated over France, both the frequency and the magnitude of snowmelt-related
events is projected to decrease (Figures 5b and 5c, and figure S5 and S6 in the supplementary
materials), while soil water excess events increase slightly. It is mainly the magnitude of the
rarest and most intense short rain and long rain events that are increasing. Marked differences
between future and historical distributions are mainly projected for events associated with non-
exceedance probabilities greater than 0.95 (i.e. floods corresponding to a 20-year return
period and beyond). However, this overall assessment at the country level is difficult to assess
and hides regional differences depending on the flood-generating processes. If we look by
region and by flood process (supplementary figures S5 and S6), we observe that this increase
in the magnitude of the rarest floods (i.e. the distribution tails), only affects the northern regions
of France, the Atlantic and continental clusters, and events linked to short rains and long rains,
which are flood events linked to rainfall on dry soils, and also the events related to soil water
excess. Under the RCP4.5, the flood distributions in the different regions for the different flood
generating process are similar between 1975-2005 and 2070-2100 (figure S5), except for
some of the most extreme flood events as mentioned above. For the RCP8.5, there are more
marked changes, notably with a shift of the flood distribution towards lower values for the
cluster corresponding to the mountainous regions, and to a lesser extent for the Mediterranean
regions. On the opposite, there are distribution shifts towards increased flood magnitudes for
most processes related to rainfall and soil water excess in the Atlantic and Continental
clusters, consistent with the overall increase in flood magnitude in these regions shown in
Figure 1.

Another important question that arises when analyzing the impact of climate change on floods
is to understand which factors have the greatest influence on flood trends in different climate
projections. Given that 11 climate projections are available here for RCP4.5 and RCP8.5, we
can calculate for each model the correlations between flood trends and the trends in the three
factors most commonly recognized as flood triggers (Wasko and Nathan, 2019, Zhang et al.,
2022) at the event scale (the antecedent soil moisture conditions, the snowmelt contribution,
and the maximal daily rainfall during the event) and the trends for floods with 2-year (Q50) and
20-year (Q95) return periods. The results of this correlation analysis are presented in Figure
8 for each of the four regions identified by the clustering of Figure 7. For the first cluster, i.e.
the mountain areas, there are strong positive correlations between snowmelt trends and Q95
flood trends, but much more contrasting correlations with Q50. For the other clusters, the
patterns are quite similar. A striking result is that the correlations between trends in floods and
trends in SWI are at least equivalent, if not superior, to the correlations with trends in intense
rainfall for the different climate projections. This result is not region-specific, and it should also
be noted that correlations with SWI are almost systematically stronger for 2-year floods, than
for 20-year floods, as already found in past observations by Wasko et al., 2021. To a lesser
extent, correlations are also weaker between trends in rainfall intensity and the flood trends
for 20-year floods, than for trends in 2-year floods. This result should however be interpreted
with caution given the uncertainties inherent in estimating intense rainfall in climate models.
These results imply that the trends projected for the various processes can, to a large extent,
explain the trends in flooding in France, and that the evolution of soil saturation plays a role
equivalent to that of the evolution of intense rainfall in the various regions to explain the trends
in flood hazard in the different climate projections.

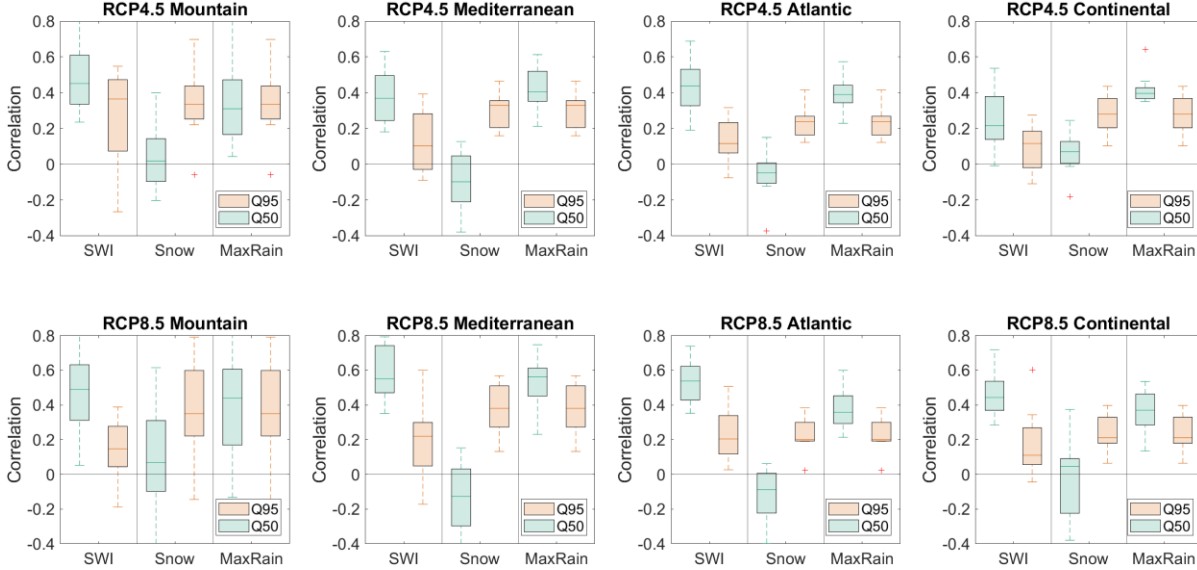

Figure 8: Correlations between the trends in event antecedent soil moisture (SWI), snowmelt
contribution (Snow) and maximum daily rainfall during the event (MaxRain), and the trends
in 2-year (Q50) and 20-year (Q95) floods, for each of the clusters presented in figure 8. The
boxplots represent the ranges of the correlation coefficients obtained for each climate
projection under the RCP4.5 (top row) and RCP8.5 (bottom row).

## 4. Discussion

These projections are in line with the expected climate change in France (Strohmenger et al., 2024): a radical change in mountainous regions with a transition to a more temperate climate, an increase in intense rainfall all over the country as in other regions of Europe (Coppola et al., 2021), and an overall North/South contrast for future rainfall changes; with an increasing trends in the northern regions, leading to higher soil moisture content, and conversely a decline in southern regions, yet with a little agreement between models (Marson et al., 2024). It is important to note that this study is one of the few (notably with Zhang et al., 2022, at the global scale) to analyze the evolution of different flood processes in future climates, rather than solely in the past as most published studies do (Tramblay et al., 2022; Tarasova et al., 2023; Kemter et al., 2023). It is interesting to note is that the conclusions of this article largely confirm that the trends and inter-variable relationships observed in the past appear to be consistent with what is happening in future simulations.

As shown in Ivancic and Shaw (2015), Tarasova et al. (2023) or Wasko and Nathan (2019), changes in extreme precipitation alone are not sufficient to explain changes in floods, and it is clearly shown in the present work that a combination of changes in flood types with changes in the frequency and magnitude of the different flood drivers best explain the temporal evolution of flood hazard. In particular, the importance of antecedent soil moisture in modulating flood hazard is expected to remain in future scenarios (as shown in Figure 8), while several observations-based studies have already shown the important role of soil moisture conditions in the genesis of floods (Penna et al., 2011; Tramblay et al., 2019, 2023), a role that is more significant for lower-magnitude floods than for more extreme floods (Wasko and Nathan, 2019; Wasko et al., 2021; Brunner et al., 2021). Given the future projections towards an increase of soil moisture droughts in large parts of Europe (Samaniego et al., 2018; Grillakis, 2019), it is expected that changes in soil moisture are key to understand changes in flood hazard in future scenarios.

For some regions, the future increase of antecedent conditions as in the central-eastern regions of France is combined with a decrease of snowmelt influence, and this was also projected for the greater Rhine basin (Rottler et al., 2021). Overall, the magnitude of snowmelt-related events is declining globally (Zhang et al., 2022) and notably in the Alps (Sikorska-Senoner and Seibert, 2020), and the present study show that the same trend is expected to continue for the future. On the contrary, given the global increasing trends in rainfall extremes, notably at short durations (Fowler et al., 2021), our projections are consistent with previous studies towards an increase of the magnitude of the rarest floods due to intense and short rain episodes in several regions and notably in the Mediterranean (Zhang et al., 2022; Tarasova et al., 2023; Poncet et al., 2025). Yet, it is worth to remind that the uncertainties on flood projections remains high notably due to the spread in climate model simulations (Evin et al., 2025).

## 5. Conclusions

Flood trends show a contrasting signal in the different regions, with a fairly good consistency towards an increase in floods in northern France and an absence of signal in southern regions. In line with the uncertainty in future precipitation changes, it should be emphasized that multi-model agreement remains fairly weak in most basins for increasing or decreasing trends in flood hazard. In terms of magnitude, trends are more marked for rare floods (20-year return period) than for frequent floods (2-year return period). Trends in initial moisture conditions show a spatial distribution quite similar to that of floods, with an increase mainly in northern regions and a decrease in antecedent soil moisture in the south. These changes tend to mimic the seasonal precipitation changes, with an increase in winter precipitation in the north and a decrease in summer precipitation in the south. The magnitude of rainfall events increases everywhere, most markedly in the northern half of France, while the contribution of snowmelt decreases everywhere in the mountain ranges, consistently with the projected decrease in snowfall as indicated by climate projections. A classification of floods by process reveals that floods linked to soil saturation account for more than half of all floods in France, and that this proportion is increasing in the north-east of the country while decreasing in the south. The relative changes in the importance of the different flood generating processes are not spatially homogeneous and varies by region. The analysis highlights that flooding trends can be decomposed into several distinct signals linked to different flooding processes, and that the multi-model uncertainty concerning these trends varies according to the process considered. Trends in soil saturation play as important a role as trends in intense rainfall in different regions to explain changes in flood hazard in the different climate simulations.

Overall, in regions where the proportion of floods linked to antecedent saturated soils (i.e Soil Water Excess floods) is decreasing, the proportion of floods linked to short or long rains is increasing, particularly in the south. Both the frequency and magnitude of floods linked to snowmelt processes are declining in areas where these processes operate, namely mountainous areas and the north-eastern regions. In contrast, the flood magnitudes associated with rainfall on dry soils (short rain and long rain floods) tend to increase, in line with the increase of maximum event rainfall in the different scenarios. The trends in the RCP8.5 scenario are both more pronounced and allow better discrimination between regions. The changes seem fairly consistent with the distribution of French climatic zones: floods tend to increase in the continental climate zone, with both an increase in rainfall intensity and soil water content, leading to more extreme floods and a increase in the number of flood events linked to soil saturation. In the temperate oceanic zone, flooding is also on the rise, but this time in connection with the increased intensity of rainfall leading to higher direct runoff but no drastic changes in the flood event types. In the Mediterranean regions, there is no marked trend in flooding, with declining soil moisture and more uncertain trends in rainfall intensity. Lastly, in mountainous regions, there are drastic changes in flood-generating conditions, with a marked drop in snowmelt and an absence of trends, or even a decline, in flooding.

In terms of perspectives for this study, there are a number of points to note. The first is that the analysis carried out here is based on a daily time step, which is unsuitable for the analysis of flash floods, which occur, for example, in Mediterranean regions with high hourly rainfall intensities. For small basins, where only until recently high-resolution convection-permitting models (Lucas-Picher et al., 2021) are able to tackle this problem by improving the modeling of small-scale convective rainfall events (Kay, 2022; Poncet et al., 2024), there is not yet a

large ensemble of model runs available to assess the uncertainties in the projections. It should be stressed out that the results obtained herein are unlikely to be transposable to flash floods, and large convection-permitting model ensembles would be required to replicate such an analysis for flash floods. Another important aspect is that this study is based on natural hydrology processes in relation to climate change. However, changes in land use (Rogger et al., 2017; Yang et al., 2021) or the construction of infrastructures, such as dams (Zahar et al., 2008; Grill et al., 2019; Blöschl, 2022b), can greatly modulate the risk of flooding. However, this type of non-climate-related change is very difficult, if not impossible, to take into account in such a large scale and multi-basin study. Nevertheless, at more local scales, and in close interaction with land-use planning and water management stakeholders, such multi-criteria analyses could be carried out to better distinguish between climatic and non-climatic influences on the evolution of flood risk.

## Data availability

All the data of this study is available freely in open access. The bias-corrected climate model simulations are available at: https://www.drias-climat.fr/. The hydrological model simulations are available at: https://www.drias-eau.fr/ . The discharge measurements over France are available at: https://www.hydro.eaufrance.fr/ . The SAFRAN database is available at: https://meteo.data.gouv.fr/datasets/donnees-changement-climatique-sim-quotidienne/

All the technical reports of the Explore2 project are available in the repository: https://entrepot.recherche.data.gouv.fr/dataverse/explore2

## Code availability

The bias-correction method for climate simulations is available at: https://github.com/yrobink/SBCK-python

The hydrological model codes are available at: https://inrae.github.io/airGRiwrm/ and https://hydrogr.github.io/airGR/

## Author contributions

YT designed the study and performed the experiments. GT, LS, ran the GRSD model simulations, LH managed the database, LC generated the climate scenarios, GE and ES contributed to the methodology and YT wrote the paper with contributions from all the authors.

## Competing interest declaration

The authors declare that they have no conflict of interest.

## Acknowledgements

This research was financed by the Explore2 project with support from the French Biodiversity Agency (OFB) and the French Ministry of Ecological Transition (MTECT). The authors would like to thank all the members of the Explore2 consortium who contributed to the creation of this unique dataset for France.

## Financial support

This research has been supported by the Office Français de la Biodiversité (Explore2 project), the Ministère de la Transition écologique et Solidaire (Explore2 project)

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
