# Peer review of "Evolution of flood generating processes under climate change in France"

_EGUsphere, 2025_

## Author Response (AR1)

**Editor comments**

**We have now received the comments of two reviewers. After my own assessment of the manuscript, the reviewer comments, and the author's responses to these comments, I agree with reviewers that there are many issues that need to be very carefully addressed. However, I also believe, after looking at the author's responses, that many of the reviewer's concerns will be addressed by including more details on the description of the methodology and by adding more information on the validation process. Though the authors refer to a paper that is under review, as the paper has not been accepted for publication, it is important to add some details in this paper (this could be done by information in the supplementary materials as suggested by the authors in their responses). Though the points listed above highlight the main issues raised by the reviewers, please consider carefully addressing all referees' comments in your revised manuscript.**

Thank you for managing our manuscript. We have carefully addressed all the reviewer comments, see below, and revised the manuscript. We complemented the methodology, by also adding the papers describing the whole project that are now online.

**Reviewer 1**

**The study performs the analysis of the future simulations a hydrological model driven by the hydrometeorological input obtained from the ensemble of global and regional bias corrected climate models with the goal to investigate changes in flood quantiles and flood generating processes in France.**

**The manuscript is timely as it addresses a groining need for understanding future flood risk under changing climatic conditions, while accounting for the model uncertainties. However, the focus of the manuscript not quite clear to me. On one side, it seems that the focus is on the uncertainty that stems from simulations of different GCM-RCM pairs (most Figures in the main manuscript are very much focused on the model agreement and not on the changes themselves). But on the other hand, the description of the results is very much focused on description of changes and trends rather than quantification of uncertainty on the projected changes. Moreover, in my opinion the study lacks validation with the past flood observations and drivers. Please find my detailed comments below.**

We thank the reviewer for taking the time to review our manuscript. We hope that our responses below will help to dispel some of the misunderstandings about the manuscript, which generally call for greater clarification of the results.

**General comments**

**The theme of the study: As mentioned above the main theme of this study is not clear to me. The figures in the main manuscript almost exclusively focus on model agreement, while the Figures in the SI do show the changes in flood magnitudes (S5 and S6), but omit the information about the effect of model uncertainty. The description of the Results (Section 3) instead very much focuses on the changes themselves rather than the model uncertainty and its implications. If the focus is on the changes themselves a more comprehensive validation with the past of observations of floods is required in my opinion (see also my comment on the lack of validation), as well as a more comprehensive presentation of the trends in the main manuscript. Instead, if the focus is on the**

**model uncertainty, I wonder how instructive is the bias correction in this case, as it essentially aims to equalize simulations across the models. Please clarify the main theme of this study and change/compliment missing aspects accordingly.**

The two aspects are very complementary, to present both whether the different models converge on the same climate change signal, and providing the actual projected changes. In fact, these two pieces of information are necessary. For example, it would be extremely dangerous to present only average projected change figures and not the number of models converging on the same signal. This way of presenting the results could indeed be misleading, if only a small number of models project an increase – or a decrease – without considering the overall agreement between the different projections. Furthermore, giving only the actual numbers of the change signals (example +10%, +20%..) could be also misleading given the uncertainties at the different levels of the modelling chain used herein. There is a much greater confidence in the results if the majority of models are providing similar projections. We better stressed these aspects in the revised manuscript.

About model validation, we present in our detailed response below the different resources that have been already published, in the frame of the Explore2 project to which the present study belongs, to assess the reliability of the models. We also provided additional validation results in the present manuscript.

The comment on the link between uncertainties and bias correction is difficult to understand and justify: climate simulations present biases (notably precipitation) that make them unusable in hydrological models without correction, at the risk of having highly biased flow simulations that may not adequately reproduce processes linked to precipitation thresholds, in particular, when rainfall is strongly biased in climate model simulations (and this is well-known since a while, see Muerth et al 2013, among others). It is not clear whether reference is being made here by the reviewer to climate or hydrological models, since it is stated that "bias correction equalizes simulations", which is true for climate simulations, but not for hydrological simulations given the different model structures and calibration strategies. Further, it should be stressed that the sample of climate model simulations selected for the Explore2 project (and the present manuscript) have been chosen to sample the different future climate scenarios of Euro-CORDEX and to be consistent with CMIP6 scenarios over France (Sauquet et al., 2025).

Furthermore, on the specific topic of uncertainties, we submitted a paper by Evin et al. in HESS (submitted June 9, and still awaiting to be published in the online discussion). This paper has enabled us to carry out a very in-depth, multi-model analysis of both climate and hydrological models, in order to distinguish between the uncertainties associated with the modeling chain and climate variability. For flood projections, and very much in line with our manuscript, the uncertainties at the scale of France remain high, but are mainly linked to the climate model and not to the hydrological model, which led to the selection of a single hydrological model in the present study, that proved to be the most efficient for floods.

Evin G., Hingray B., Thirel G., Ducharne A., Strohmenger L., Corre L., Tramblay Y., Vidal J-P., Bonneau J, Colleoni F., Gailhard J., Habets F., Hendrickx F., Héraut L., Huang P., Le Lay M., Magand C., Marson P., Monteil C., Munier S., Reverdy A., Soubeyroux J-M., Robin Y., Vergnes J-P., Vrac M., Sauquet E., Uncertainty sources in a large ensemble of hydrological projections: Regional Climate Models and Internal Variability matter, Hydrology and Earth System Sciences, submitted.

Muerth, M. J., Gauvin St-Denis, B., Ricard, S., Velázquez, J. A., Schmid, J., Minville, M., Caya, D., Chaumont, D., Ludwig, R., and Turcotte, R.: On the need for bias correction in regional climate

scenarios to assess climate change impacts on river runoff, Hydrol. Earth Syst. Sci., 17, 1189–1204, https://doi.org/10.5194/hess-17-1189-2013, 2013.

**Lack of Discussion: Regardless of the choice of the topic, the Discussion is currently completely absent in this study. The section Results and Discussion provides only very few references to the previous studies. Putting own results in the context of previous research (either in regard of model uncertainty or in regard of explaining future changes in flood quantiles and drivers) would be extremely beneficial for this manuscript and can help to clarify the implications of the predicted flood changes.**

We agree, the reason why this discussion part was perhaps a little short is due to the fact that a very large volume of results had to be presented in this manuscript. We have improved the discussion, as also requested also from Reviewer 2.

**Lack of Validation: Apart from a few notes in text that the simulation results are consistent with previous studies, I do not see any validation with the past observed floods. If the focus of the manuscript is indeed purely on the model uncertainty for future simulations, I find it acceptable, but since most of the results explained in the text rather refer to the future changes in flood quantiles compared to the past, the reliability of simulations for the past periods with existing observations have to be quantitatively proven.**

The simulations used in the present study originate from the Explore2 project, and the full database is already presented, in particular the validation of the model, in the following paper currently under review in HESS:

Sauquet E., Evin G., Siauve S., Aissat R., Arnaud P., Berel M., Bonneau J., Branger F., Caballero Y., Colleoni F., Ducharne A., Gailhard J., Habets F., Hendrickx F., Heraut L., Hingray B., Huang P., Jaouen T., Jeantet A., Lanini S., Le Lay M., Magand C., Mimeau L., Monteil C., Munier S., Perrin C., Robelin O., Rousset F., Soubeyroux J.-M., Strohmenger L., Thirel G., Tocquer F., Tramblay Y., Vergnes J.-P., Vidal J.-P., A large transient multi-scenario multi-model ensemble of future streamflows and groundwater projections in France, Hydrology and Earth System Sciences, https://egusphere.copernicus.org/preprints/2025/egusphere-2025-1788/

In table 3 of this paper, it can be shown that several metrics dedicated to extremes have been considered for the validation the hydrological models: Q10, the relative error of the 10% maximum discharge, medtQXJA, the bias on the median occurrence (julian day) of the annual maximum daily streamflow, in addition to the 14 other metrics used for the validation of the models. In table 5 of Sauquet et al., it can be seen that the GRSD model is the one with the smallest errors on high flow in the largest number of basins, so this lead us to choose this model for the present study.

In addition, the results for all singles stations are available online (the global report on model performance, 132 pages, https://doi.org/10.57745/S6PQXD, and the individual summary sheet for all stations : https://doi.org/10.57745/LNTOKL ). As noted in the data availability section, all the data of the Explore2 project are 100% open access.

But to answer this criticism of "lack of validation" we have completed this validation specifically for the extremes used in this article and added these new results in supplementary materials. See this new results in our response to the specific comment below.

**Specific comments**

**Line 61: Floods with 20-year return period are hardly the "rarest", please be more specific.**

We changed to 20-year floods.

**Line 101: I do not think that the studies of Froidevaux et al., 2015, Tarasova et al., 2018 and Kim et al., 2019 are relevant here, as they do not directly examine different flood types. I think the remaining 6 reference suffice.**

We removed Froidevaux et al., 2015, Tarasova et al., 2018 and Kim et al., 2019.

**Line 125-131: The purpose of this statement is not quite clear to me. Is it supposed to be the justification for the absence of validation? Regardless of the chosen fashion of model implementation (i.e., with or without calibration), validation is an essential part of any simulations study regardless of the choice of modeling chain configuration (even if the hydrological model allegedly plays minor role in the overall uncertainty). Please see also my major comments.**

No, it is no justification for the absence of validation. In this section we refer to the literature about published studies on climate change impacts on floods. As a matter of fact, many large-scale studies are using uncalibrated models over large domains. We agree that validation is important, and the model simulations used in the present study have been validated, see our previous comment.

**Line 230-232: Not quite clear how the baseflow contribution was estimated. Was it done for a single day that corresponds to the peak discharge of annual maximum flood? Please clarify.**

The base flow filter was applied to the full time series to extract for each day the base flow component and the direct runoff component. So, for the days corresponding to floods we can estimate the direct runoff contribution to the total runoff. We better explained this in the revised document.

**Line 233-235: It might be warranted to search for the precipitation within 7 days before the start of the event to account for the time of concentration given the size of the study catchments, but it provides little justification for such search window relative to the flood peak, as in this case such search window represents not the time of concentration but the duration of corresponding rainfall or snowmelt event. This could be especially a limiting factor for snowmelt events, where build up time can be particularly long. Hence this potentially can lead to misclassification between "Rain and snowmelt events" and the rainfall events. A designated sensitivity analysis might shed light on the effect of the window search on the classification results.**

We would like to stress that this search window of 7 days is fully consistent with many published studies related to floods. In Ivancic and Shaw, 2015, they analyze the link between rainfall and floods in 390 basins and found that 93% of sites had a lag between precipitation and discharge less than 5 days. Stein et al 2019 also used a search window of 7 days in 4155 catchments worldwide. In Berghuijs et al. (2016) for 420 catchments in the United States that, it varies between 3 and 10 days, so they decided to set it to 7 days. Do et al. 2020 also used 5 days for 671 catchments across the USA.

In addition, we tested in our methodology a time window of 30 days instead of 7 days. See the results below. In Figure R1 you can see the different flood event classes, with a very similar classification as when using a maximum 7-day search window. The snowmelt event represents a small proportion at the scale of France, 3.73% if considering 30 days and 3.08% if considering 7 days as in our previous results. Similarly, the Rain and snowmelt event account for 7.60% of event when considering 30 days, 7.40% when considering 7 days. In figure R2, the changes in the different flood generation mechanisms are plotted, using a 30-day maximum search windows, with results almost identical to the results presented in the manuscript.

Consequently, we kept the 7-day time window.

Do, H. X., Mei, Y., & Gronewold, A. D. (2020). To what extent are changes in flood magnitude related to changes in precipitation extremes? Geophysical Research Letters, 47, e2020GL088684. https://doi.org/10.1029/2020GL088684

[Figure]

Figure R1: Flood event classification, using a maximum time window of 30 days, instead of 7 days, to classify floods.

[Figure]

Figure R2: Change in the frequency of flood types, using a maximum time window of 30 days to classify the flood events, instead of 7 days.

**Line 262-269: Some of these classification rules are rather ambiguous. Rule 1: it is not clear if volumes or rates are compared here. In case volumes are compared, it is not clear for which periods they are computed. Rule 4 and 5: it is not clear if these distinctions are only done when soil moisture threshold is not exceeded. Please clarify.**

The classification used herein has also been used by Kemter et al 2023, and is very similar to the classification of Stein et al 2020.

Rule 1: the snowmelt is in mm, as the rainfall, so the two quantities are comparable. As explained in the methodology lines 240-243: "For each simulated flood event, we extracted: the antecedent soil moisture (i.e. SWI, between zero and one) one day before the flood date, the concentration time (in days), the total and maximum daily rainfall (mm) during the event, the fraction of the total flood streamflow being direct streamflow (%), and the total snowmelt discharge (mm) during the event. "

For Rules 4 and 5, as noted line 262, these are sequential rules (corresponding to a elseif instruction in a loop), so if rule 4 is valid, it means that the precedent rules are not valid. The same for the other rules. Rules 4 and 5 are corresponding to rainfall events when the soil moisture threshold is not exceeded.

**Line 276-278: It is not clear how streamflow events were identified here, or are these streamflow rates? Are these relationships obtained from the observed or from the modelled data? Please clarify. Moreover, please clarify why particularly this threshold is expected to be appropriate for distinguishing "Soil water excess" floods.**

We use here the same approach as Tarasova et al., 2018; Tramblay et al., 2022. It is based on the extraction of all runoff days above the 10th percentile, to check the relation with antecedent soil moisture, and then identify an inflexion point in this relation indicative of saturated soil conditions. Since the whole paper is on model-based simulations of river discharge, this approach is used for each model simulation. As shown in supplementary figure S1, the SWI threshold of 0.75 seems to be the most adapted in the case of France. It is interesting to stress that the exact same threshold was also identified by Kemter et al 2023 on a totally different set of catchments in the US.

**Line 295-297: Although this study aims to analyze trends in the 2 and 20-year return period floods, the performance of the model chain is not shown neither for representing past floods with different return periods nor for representing their changes in the past. Please also see my major comment.**

We agree. Even if the references of Sauquet et al. 2024, 2025, are presenting a complete validation of the modelling chain on many indicators indicating mean flow, extremes… we added the comparison of the 2-year (Q2) and 20-year (Q20) quantiles between the observations and the hydrological model. Please see the figure R3 below. As shown on figure R3, the mean error is equal to 16% for Q2 and 20% for Q20, that could be considered low given that we are dealing here with extreme values. We added this figure in supplementary materials.

About representing changes in flood in the past, it is not feasible given the short time period (1975-2005) that prevent a robust assessment. Climate models that are not driven by reanalysis data but CMIP simulations are not able to reproduce an accurate day-to-day chronology (given the impact of internal variability and model initialization), so they are not fit for the purpose of identifying past trends (Randall et al. 2007, Scinocca et al., 2015), notably on short time periods (this is why in the present manuscript we do the trend detection on the full time series 1975-2100).

Scinocca, J. F., Kharin, V. V., Jiao, Y., Qian, M. W., Lazare, M., Solheim, L., Flato, G. M., Biner, S., Desgagne, M., & Dugas, B. (2015). Coordinated Global and Regional Climate Modeling*. Journal of Climate, 29(1), 17–35. https://doi.org/10.1175/jcli-d-15-0161.1

Randall, D.A., et al., 2007. Climate models and their evaluation. Chapter in: Climate change 2007: The physical science basis. Contribution of Working Group I to the Fourth Assessment Report of the Intergovernmental Panel on Climate Change. Cambridge University Press. Available at: http://ipcc-wg1.ucar.edu/wg1/wg1-report.html

[Figure]

Figure R3: Comparison of observed and simulated extreme river discharge corresponding to a 2-year (Q2) and 20-year (Q20) return levels. For the computation of the quantiles, the annual maximum time series have been extracted during the period 1975-2005 and fitted with a GEV distribution with the L-Moments approach.

**Line 302: Please clarify what is meant here by flood event characteristics**

We added "(event rainfall, antecedent soil moisture, direct runoff fraction, snowmelt contribution)"

**Line 306: I wonder if minimum 20 occurrences per process is sufficient for robustly detecting trends in floods with 20 year-return period.**

It is not written in line 306 that we used 20 occurrences to detect changes in 20-year floods. We used a minimum of 20 occurrence to test for trends in the frequency of the different flood event types. We better stressed in the text how the trend detection was carried out, on which variables.

**Line 319: Please explain the acronym before using it.**

If you refer to MIA, line 318, it is already defined line 308.

**Figure 1: I have difficulties interpreting the agreement between the models, because the meaning of the multi-model agreement index is not explained anywhere in the text and because it is not clear to me how this Figure and Figure S2 are connected and if they show the same information.**

**Please clarify and make sure that all captions provide all the information needed for interpreting the corresponding Figures.**

Actually, in section 2.5, lines 307-310 the MIA index is described as: "To present the trend detection results, we considered the multi-model index of agreement (MIA) (Tramblay and Somot, 2018) that describes the model convergence towards an increase (1) or a decrease (-1) for a given indicator. The objective is to identify robust changes, where all model projections converge towards the same result." So, it is not correct to state that the multi-model agreement index is not explained anywhere in the text.

Figure 1 show the value of the MIA for each basin. Figure S2 show the boxplot of the distribution of this MIA index over France. We improved the captions.

We added in Figure 1 a repetition of lines 308-309: "The multi-model index of agreement (MIA) is equal to 1 if all models project a significant increase, and equal to -1 if all models project a significant decrease. "

**Line 350: Please clarify what is direct runoff in this case and how it is defined. Please also provide this information in the caption of Figure 3.**

We added "that is the total runoff minus the base flow for the day of the flood event". We did not include it again in the figure caption, since it would be redundant and it is a quite standard variable in hydrology.

**Line 357: Where these trends are displayed? It is difficult to judge the accuracy of such statements since the spatial trends are not displayed in any of the figures, only model agreement about the trends.**

Figure 3. We added "seen in Figure 3". The spatial trends are shown on figure 3, not in terms of absolute value but the proportion of models showing a significant upward or downward trend. This is way more robust to present results this way, see our previous comments.

**Line 396-408: I miss here the quantitative comparison with the flood classification using past observations in France. Please see also my major comment about the lack of validation.**

We provided the comparison of the classifications, using the different bias-corrected simulations during the historical period. See Figure S4, showing that the different bias-corrected models are able to reproduce basically almost the same distribution of flood types, with the relative proportion of the different classes fully consistent with the literature for Western Europe/ France. This is somewhat expected, given that all climate models are corrected with the same data, the SAFRAN-ISBA reanalysis over France. Given that, after bias correction, the model simulations during the historical period 1975-2005 are matching the "observations" (in fact, a reanalysis) it is fully equivalent to show the classification with the "observations". This is proven by Figure S4, with all bars basically the same across models.

**Figure 5: It is rather difficult to judge the differences between there panels, correspondingly making it rather difficult to see the changes. Consider displaying these results in a different way. Please also clarify in the caption the meaning of the values on the x axes.**

We updated the figure 5, in order to have the same x axis. The changes in flood magnitude for the different flood generating processes are already shown in a different way, in figure S5 and S6 (using kernel distributions), that are by far easier to interpret to assess changes in flood magnitude. The figure 5 is mostly useful to show the difference in magnitude between the different types of floods,

whereas figures S5 and S6 aim at documenting the change in flood intensity for each process in future scenarios. The two figures are therefore complementary. We agree that the reference to figure 5 only was not sufficient so we added: "The magnitude of floods will change differently, depending on their generating processes. When aggregated over France, the magnitude of snowmelt-related events is projected to decrease (Figures 5b and 5c, and figure S5 and S6 in the supplementary materials), while soil water excess events increase slightly".

We added in the caption "The annual flood (AMF) for each station have been standardized by the mean annual flood ($\overline{AMF}$)"

**Line 452-455: This part sounds like a description of the Methods. Please move it to the corresponding section.**

We moved this part into the methods.

**Figure 7: Please provide a detailed description of each cluster in the caption. Without it this figure is not very helpful for the readers.**

Thanks for this comment, that will indeed improve the readability of the results. To be consistent, we changed the cluster names for geographic regions, in this figure and the rest of the document.

**Line 480-482: These changes are very difficult to see from that Figure. See also my comment to Figure 5.**

We replied before. We have a much clearer view of these changes in Figures S5 and S6.

**Line 493-494: It is not quite clear which extremes are meant here. Please clarify.**

Some of the most extreme events, changed to "some of the most extreme flood events".

**Line 493-498 and elsewhere in this Section: I miss here completely a discussion about the anticipated changes, their possible causes and agreement with anticipated changes with other hydrometeorological variables. See also my main comment ton the lack of discussion.**

We agree and we included a whole new section of discussion:

"These projections are in line with the expected climate change in France (Strohmenger et al., 2024): a radical change in mountainous regions with a transition to a more temperate climate, an increase in intense rainfall all over the country as in other regions of Europe (Coppola et al., 2021), and an overall North/South contrast for future rainfall changes; with an increasing trends in the northern regions, leading to higher soil moisture content, and conversely a decline in southern regions, yet with a little agreement between models (Marson et al., 2024). It is important to note that this study is one of the few (notably with Zhang et al., 2022, at the global scale) to analyze the evolution of different flood processes in future climates, rather than solely in the past as most published studies do (Tramblay et al., 2022; Tarasova et al., 2023; Kemter et al., 2023). What is interesting to note is that the conclusions of this article largely confirm that the trends and inter-variable relationships observed in the past appear to be consistent with what is happening in future simulations. As shown in Ivancic and Shaw (2015), Tarasova et al. (2023) or Wasko and Nathan (2019), changes in extreme precipitation alone are not sufficient to explain changes in floods, and it is clearly shown in the present work that a combination of changes in flood types with changes in the frequency and magnitude of the different flood drivers best explain the temporal evolution of flood hazard. In particular, the importance of antecedent soil moisture in modulating flood hazard is expected to remain in future scenarios (as shown in Figure 8),

while several observations-based studies have already shown the important role of soil moisture conditions in the genesis of floods (Penna et al., 2011; Tramblay et al., 2019, 2023), a role that is more significant for lower-magnitude floods than for more extreme floods (Wasko and Nathan, 2019; Wasko et al., 2021; Brunner et al., 2021). Given the future projections towards an increase of soil moisture droughts in large parts of Europe (Samaniego et al., 2018; Grillakis, 2019), it is expected that changes in soil moisture are key to understand changes in flood hazard in future scenarios. For some regions, the future increase of antecedent conditions as in the central-eastern regions of France is combined with a decrease of snowmelt influence, and this was also projected for the greater Rhine basin (Rottler et al., 2021). Overall, the magnitude of snowmelt-related events is declining globally (Zhang et al., 2022) and notably in the Alps (Sikorska-Senoner and Seibert, 2020), and the present study show that the same trend is expected to continue for the future. On the contrary, given the global increasing trends in rainfall extremes, notably at short durations (Fowler et al., 2021), our projections are consistent with previous studies towards an increase of the magnitude of the rarest floods due to intense and short rain episodes in several regions and notably in the Mediterranean (Zhang et al., 2022; Tarasova et al., 2023; Poncet et al., 2025)."

And the new references herein have been added to the revised manuscript.

**Line 506: Please clarify if rainfall rate or rainfall volume was meant here.**

We added: **"**the maximal daily rainfall during the event"

**Line 516: More frequent floods?**

We changed to "2-year floods"

**Line 507-522: It would be very helpful to put these results into perspective. See my comment above.**

Same response as above, we added a discussion.

**Figure 8: Please add explanation of clusters in the caption. Please also clarify what is meant here by maximum rainfall. Event? Rate? Volume?**

We added: "and maximum daily rainfall during the event". We also refer to the figure 7, for the description of the clusters. As mentioned before, we changed the cluster names to be consistent.

**Figure S2: Please clarify what MIA stands for and how its values can be interpreted.**

Already explained in the methodology.

**Figure S3: It would be more instructive to indicate not the absolute SWI values that were reached during the Long events, but instead to indicate how often was the threshold that splits between Long events and Soil excess events exceeds.**

Thanks for this suggestion, that is indeed a useful complementary result. As explained in section 2.4, this threshold is basin dependent. So, we computed this proportion for each basin separately, on average, for 69% of long rain floods, the threshold that splits between long rain and soil excess floods are exceeded. We added this new result to the manuscript.

**Figure S4: It is difficult to judge visually agreement between the bars. Please provide a qualitative metrics that supports the statement that models consistently represent flood generation processes of the past. I repeat here my earlier comment again, it would be very useful to compare the**

**consistency of these classifications with the classifications based on observations during the historical period.**

In addition to the plot of figure S4, here is below a table giving the actual numbers, for each climate simulation during the historical period. Given that these simulations are bias-corrected with the same reference, it is expected that very similar results will be obtained, as shown on the table below.

| Climate model | | Flood generating process between 1975-2005 | | | | |
|---|---|---|---|---|---|---|
| **General Circulation Model** | **Regional Climate Model** | **Snowmelt** | **Rain and snowmelt** | **Soil Water Excess** | **Short rain** | **Long rain** |
| CNRM-CERFACS-CNRM-CM5 | CNRM-ALADIN63 | 3% | 6% | 55% | 9% | 28% |
| CNRM-CERFACS-CNRM-CM5 | KNMI-RACMO22E | 3% | 8% | 50% | 10% | 29% |
| ICHEC-EC-EARTH | KNMI-RACMO22E | 4% | 7% | 52% | 9% | 28% |
| ICHEC-EC-EARTH | SMHI-RCA4 | 3% | 8% | 56% | 10% | 23% |
| MOHC-HadGEM2-ES | CLMcom-CCLM4-8-17 | 3% | 7% | 51% | 11% | 27% |
| IPSL-IPSL-CM5A-MR | SMHI-RCA4 | 3% | 7% | 57% | 10% | 23% |
| IPSL-IPSL-CM5A-MR | IPSL-WRF381P | 2% | 8% | 57% | 7% | 26% |
| MPI-M-MPI-ESM-LR | CLMcom-CCLM4-8-17 | 3% | 8% | 54% | 9% | 26% |
| MPI-M-MPI-ESM-LR | MPI-CSC-REMO2009 | 3% | 8% | 52% | 10% | 26% |
| NCC-NorESM1-M | DMI-HIRHAM5 | 3% | 7% | 59% | 9% | 22% |
| NCC-NorESM1-M | GERICS-REMO2015 | 4% | 7% | 54% | 9% | 25% |

**Figure S5: Please name the clusters based on the corresponding geographical areas and indicate it in the caption, this would very much support interpretability of this figure, as well as the figures in the main manuscript.**

We have updated the figure, and changed the names of the clusters.

**Reviewer 2**

**The manuscript "Evolution of flood-generating processes under climate change in France" presents an analysis of trends in flood magnitudes and flood-generating processes for 3727 catchments in France. Using a hydrological model and 22 climate simulations, flood events were predicted for the time period 1975 to 2100. The authors find four distinct clusters of future flood changes in France. Flood magnitudes in the North are increasing. The driver in the North-East is a decrease in snowmelt floods and an increase in soil water excess floods. The driver in the North-West is less clear. The authors attribute the change to an increase in short rain floods, though the model agreement for this change is low (Figure 6). In the South, flood trends and drivers can be clustered into mountainous regions with a decrease in flood magnitude, explained by a decrease in snowmelt-influenced floods and an increase in short rain/long rain floods. The other cluster in the South collects the Mediterranean catchments, which show no trend or a decreasing trend in magnitude, explained by a decrease in saturation excess floods and an increase in short rain and long rain floods.**

**Overall, I find distinguishing future flood development by generating process highly relevant, and the method of analysis very good. However, the reason I summarised the four clusters, is that the manuscript could benefit from more clarity and consistency in describing the trends, drivers, and clusters. At the moment, there are some statements that contradict each other, or contradict the results shown in the figure (see detailed explanation below). Since I find the underlying method, analysis, and figure to be sound, I am sure the authors will be able to easily address these points.**

Thanks for this positive and constructive evaluation of the manuscript. We addressed the different issues below.

**Specific comments:**

**The statements regarding trends in the contribution of short rain floods are not consistent between the abstract and the results/figures. In the abstract, it says:**

**"The proportion of floods linked to soil saturation excess is decreasing while the proportion of floods linked to infiltration excess related to extreme rainfall is increasing, particularly in the southern Mediterranean regions."**

**Whereas in Section 3.3 and Figure 6, it is clearly discussed that the strongest increase in most catchments is the increase in saturation excess floods, with the exception of the Mediterranean region, which only makes up a small number of catchments compared to the catchments in the North.**

Thank you for noticing that, we realized that the abstract was an old version not updated with the final results.

The sentence: "The proportion of floods linked to soil saturation excess is decreasing while the proportion of floods linked to infiltration excess related to extreme rainfall is increasing, particularly in the southern Mediterranean regions", has been replaced by:

"The proportion of floods linked to soil saturation excess is increasing in the Northeast, while decreasing in the southern Mediterranean regions. In these Mediterranean regions, the frequency of floods linked to infiltration excess related to extreme rainfall is increasing."

**Since there is quite a lot of information being presented (trends in magnitude, trends in drivers, trends in process frequency, correlations between drivers and magnitude) that all contribute to explaining changes in flooding, I would recommend increasing clarity by:**

1. **Being very clear about if a mentioned increase/decrease/trend is referring to changes in magnitude, or frequency.**

2. **Use consistent terms for the flood drivers (e.g. infiltration excess vs rainfall on dry soil, soil water excess vs saturation excess)**

3. **Giving the clusters more descriptive names as well as showing how many catchments are included in each cluster. In Section 3 the clusters are referred to only by number, but then in the conclusion, they are suddenly only mentioned by regions (Mountainous, Mediterranean, North, temperate oceanic zone…). I find these descriptions actually more helpful and would recommend adopting them earlier, together with a visual summary of what the distinguishing aspects of each cluster are.**

We agree. We clarified everywhere it was needed the trends in magnitude and/or frequency, in the methods and result sections. About your comment above that "The driver in the North-West is less clear. The authors attribute the change to an increase in short rain floods, though the model agreement for this change is low (Figure 6)", we better explained the results. For this region, there is indeed not a sharp increase in the frequency of short rain floods (figure 6). But from figure S5 and S6, it can be seen a increase in the magnitude of the event associated with Soil water excess, Short rain and long rain floods, fully in line with increase in heavy rainfall (Figure 3), that can likely explain the increase in flood magnitude in this region. This is indeed a good example of the interplays between changes in the flood event frequency and flood event magnitudes. We modified a bit the conclusion to stress this point.

We also improved the consistency of the terms describing floods drivers, notably to clearly assign to a given class the corresponding process at play (ie. soil infiltration excess vs. soil saturation excess).

We changed the names of the clusters; indeed, it is a good idea to use more descriptive names by the regions, and include the number of catchments in each region. We updated figures 7 and 8.

**Further comments:**

**Basins located in Cluster 1 show a clear increase in short rain and long rain flood contributions (Figure 6). Why do the flood magnitudes decrease (Figure 1), if particularly short rain floods are associated with higher magnitude events (Figure 5, line 568/569)?**

The basins in cluster 1 are the mountainous regions of the Alps and the Pyrenees. In these basins, the dominant processes are snowmelt and rain and snowmelt floods, as shown in Figure 4. So, given that there is a sharp reduction of the intensity of floods related to these processes, this is the reason why the flood magnitude is decreasing, when considering all processes together as in Figure 1.

**It is unclear, both from the abstract and the methods section, if the model simulations are part of the work done for this analysis, or if they have been published previously. From the cited reference Tramblay et al, 2024, I would assume the latter. Please make it clear, that existing simulations are being used. Otherwise, I would expect more model evaluation to be part of this manuscript.**

The simulations used in the present study originate from the Explore2 project, and the full database is already presented, in particular the validation of the model, in the following paper currently under review in HESS:

Sauquet E., Evin G., Siauve S., Aissat R., Arnaud P., Berel M., Bonneau J., Branger F., Caballero Y., Colleoni F., Ducharne A., Gailhard J., Habets F., Hendrickx F., Heraut L., Hingray B., Huang P., Jaouen T., Jeantet A., Lanini S., Le Lay M., Magand C., Mimeau L., Monteil C., Munier S., Perrin C., Robelin O., Rousset F., Soubeyroux J.-M., Strohmenger L., Thirel G., Tocquer F., Tramblay Y., Vergnes J.-P., Vidal J.-P., A large transient multi-scenario multi-model ensemble of future streamflows and groundwater projections in France, Hydrology and Earth System Sciences, https://egusphere.copernicus.org/preprints/2025/egusphere-2025-1788/

In table 3 of this paper, it can be shown that several metrics dedicated to extremes have been considered for the validation the hydrological models: Q10, the relative error of the 10% maximum discharge, medtQXJA, the bias on the median occurrence (julian day) of the annual maximum daily streamflow, in addition to the 14 other metrics used for the validation of the models. In table 5 of Sauquet et al., it can be seen that the GRSD model is the one with the smallest errors on high flow in the largest number of basins, so this led us to choose this model for the present study.

In addition, the results for all singles stations are available online (the global report on model performance, 132 pages, https://doi.org/10.57745/S6PQXD, and the individual summary sheet for all stations : https://doi.org/10.57745/LNTOKL ). As noted in the data availability section, all the data of the Explore2 project are 100% open access.

We updated the text to explain these points. We also added a validation specific to the 2-year and 20-year quantiles used in the present study.

**In regard to the flood process classification, how influential is it, that the antecedent soil moisture is extracted the day before the flood event? Kemter et al (2020, 2023) extract soil moisture the day before time of concentration. Similar for Tarasova et al (2020) where antecedent moisture is extracted before the separated runoff event, and Stein et al (2020) who extracted antecedent moisture 7 days before the flood event. Since the time of concentration based on rainfall distribution is extracted anyway, why not extract antecedent moisture the day before the rainfall event? Otherwise, antecedent moisture will have already been influenced by the rain.**

It is a mistake in the text, thank you for spotting it!

We use the exact same approach as in Kemter et al or Tarasova et al. We changed the sentence to: "For each simulated flood event, we extracted: the antecedent soil moisture (i.e. SWI, between zero and one) one day before the concentration time…".

**In line 514 you discuss that "correlations with SWI are almost systematically stronger for frequent floods (2-year), than for rarer floods (20-year)." Please discuss how that matches previous findings that frequent floods are more likely to be soil moisture influenced than rare floods (e.g. Wasko et al, 2021).**

We added the reference, and also a whole new paragraph at the end of this section to better discuss the results in relation with previous literature.

**Technical comments:**

**L60: What does "The trends in flood magnitude are contrasted, with increasing trends only in the northern regions of France" mean? Currently the sentence does not make sense.**

We replaced by: "Increasing trends in flood magnitudes are only found in the northern regions of France, although multi-model convergence rarely exceeds 60 %."

**L118: space missing**

Fixed, thank you.

**Figure 5: Please keep the x axis consistent between the three plots.**

Thank you for spotting this, we changed the plot.

**Figures: Please ensure for all figures that colors are accessible for people with color vision deficiency (Stoelzle & Stein et al, 2021). This is particularly and issue in Figure 5 and 7**

Thank you, we have updated both figures. All images have been created using colors from https://colorbrewer2.org and verified using https://www.color-blindness.com/coblis-color-blindness-simulator/ and https://bioapps.byu.edu/colorblind_image_tester (and verified by one of the co-author with color vision deficiency).

**L568: The correct term should be "In contrast, …" not "On the contrary, …".**

Changed.

**L568: Missing letter in "soil"**

Changed.

**Stoelzle, M., & Stein, L. (2021). Rainbow color map distorts and misleads research in hydrology–guidance for better visualizations and science communication. Hydrology and Earth System Sciences, 25(8), 4549-4565.**

**Wasko, C., Nathan, R., Stein, L., & O'Shea, D. (2021). Evidence of shorter more extreme rainfalls and increased flood variability under climate change. Journal of Hydrology, 603, 126994.**